# Assessing Healthcare Professionals’ Identification of Paediatric Dermatological Conditions in Darker Skin Tones

**DOI:** 10.3390/children9111749

**Published:** 2022-11-15

**Authors:** Dhurgshaarna Shanmugavadivel, Jo-Fen Liu, Danilo Buonsenso, Tessa Davis, Damian Roland

**Affiliations:** 1Academic Unit of Lifespan and Population Health, School of Medicine, University of Nottingham, Nottingham NG7 2UH, UK; 2Department of Woman and Child Health and Public Health, Fondazione Policlinico Universitario A. Gemelli IRCCS, 00168 Rome, Italy; 3Dipartimento di Scienze Biotecnologiche di Base, Cliniche Intensivologiche ePerioperatorie, Università Cattolica del Sacro Cuore, 00168 Rome, Italy; 4Global Health Research Institute, Istituto di Igiene, Università Cattolica del Sacro Cuore, 00168 Roma, Italy; 5Paediatric Emergency Department, Royal London Hospital, London E1 1BB, UK; 6Blizard Institute, Queen Mary University of London, London E1 2AT, UK; 7Paediatric Emergency Medicine Leicester Academic (PEMLA) Group, Children’s Emergency Department, Leicester Royal Infirmary, Leicester LE1 5WW, UK; 8SAPPHIRE Group, Health Sciences, Leicester University, Leicester LE1 7RH, UK

**Keywords:** child health, paediatrics, medical education, dermatology, ethics

## Abstract

The impacts of the lack of skin tone diversity in medical education images on healthcare professionals (HCPs) and patients are not well studied. The aim of this study was to assess the diagnostic knowledge of HCPs and correlate this with confidence and training resources used. An online multiple choice quiz was developed. The participants’ demographics, training resources and self-confidence in diagnosing skin conditions were collected. The differences in the results between the subgroups and the correlations between the respondents’ experience, self-reported confidence and quiz results were assessed. The mean score of 432 international participants was 5.37 (SD 1.75) out of a maximum of 10 (highest score). Eleven percent (n = 47) reached the 80% pass mark. Subanalysis showed no difference by the continent (*p* = 0.270), ethnicity (*p* = 0.397), profession (*p* = 0.599), training resources (*p* = 0.198) or confidence (*p* = 0.400). A significance was observed in the specialty (*p* = 0.01). A weak correlation between experience and confidence (Spearman’s ρ = 0.286), but no correlation between scores and confidence or experience (ρ = 0.087 and 0.076), was observed. Of diagnoses, eczema was recognised in 40% and meningococcal rash in 61%. This is the first study assessing the identification of paediatric skin conditions in different skin tones internationally. The correct identification of common/important paediatric conditions was poor, suggesting a possible difference in knowledge across skin tones. There is an urgent need to improve the representation of all skin tones to ensure equity in patient care.

## 1. Introduction

Equality, diversity and inclusion are regarded as crucial topics of conversation globally, with a call for equitable representation to address the healthcare needs of a rapidly changing diverse population [1]. In the UK, racism engrained in the medical culture needs urgent attention, and more recently, there has been a call to focus on medical education. Despite formal curriculums advocating for equality and diversity, the clinical need is far from being met. The use of images within medical education are crucial and aid in the recognition of key clinical conditions. However, there is a real lack of diversity in the representation of wide-ranging skin tones within medical education [2]. Studies have shown that more than 74% of images within textbooks, scientific journals and teaching slide decks feature images of light skin tones [3,4,5,6]. The recent pandemic has further highlighted inequities, with a systematic review reporting that the cutaneous manifestations of COVID-19 reported in the literature did not include a single image of darker skin tones [7]. This was despite clear evidence that COVID-19 disproportionately affected those from ethnic backgrounds [1].

This lack of representation has an impact on patient care, with delays in recognition and treatment. These health disparities are well documented within dermatology, with non-White patients having higher morbidity and mortality than their White counterparts [8]. Medical students lack specific dermatology teaching in darker skin tones and are not as proficient in diagnosing skin conditions in darker skin tones [9,10]. At the postgraduate level, there is a desire for more specific skin tone teaching, with 75% of Australian dermatologists expressing a lack of confidence in performing procedures on skin of colour and 80% stating that they would have liked more specific teaching of darker skin tones during their training [11].

Whilst the poor diversity is well documented, there is a dearth of data on the impacts of this on the confidence and knowledge of healthcare professionals (HCPs) and patient outcomes. Within paediatrics, this has been a particular concern and led to the development of a global online project called Skin Deep, collecting paediatric images in a range of skin tones for use in medical education by both HCPs and the public [12]. The team previously explored the diversity of resources used when training healthcare professionals and their self-rated confidence in diagnosing paediatric dermatological conditions in children and young people of all skin tones [13]. The aim of this study was to assess the diagnostic knowledge of HCPs using paediatric images of skin conditions in darker skin tones and to identify whether this correlated with confidence and the training resources used.

## 2. Methods

### 2.1. Quiz and Scoring

An online platform had previously been developed by the not-for-profit Don’t Forget the Bubbles (DFTB) Skin Deep team as part of a global collaboration (www.dfbskindeep.com last accessed on 13 November 2022). The website hosts appropriately consented high-quality images of paediatric skin and genetic conditions across all skin tones.

As part of the purpose-built “Quiz” section of the website, participants were required to login and enter their demographic details, which included ethnicity, profession, specialty, continent of practice, training resources used while training (i.e., white skin, a mix of skin tones and darker skin tones only), and self-confidence in diagnosis across a range of skin tones. The self-reported confidence was based on one of three categories: generally uncertain if correct, sometimes uncertain but clinically safe, and confident across range of skin tones. Participants were informed that their demographic information would be shared as part of a research project prior to inputting their data.

Participants were then able to access the quiz, which consisted of a series of 10 images with four possible answers and asked which condition was depicted in the image (Figure 1). The questions were randomly generated from a library of 81 images featuring a total of 56 diagnoses (some diagnoses had more than one image), including chickenpox, abscesses, burns, warts, café-au-lait macule scrofuloderma, systemic lupus eritematous and others, as detailed in Appendix A. Of these images, 43 were dark, 23 medium and 15 light skin tones, all having previously been collected from clinical practice settings with written consent from families and reviewed by a team of dermatologists. The questions were randomly generated and not the same for all participants, as we felt due to the open nature of the website it might be possible for participants to easily share answers. The pass mark for the quiz was set at 80%.

The project was deemed not to require formal ethical approval as per the Health Research Authority (HRA) assessment tool.

### 2.2. Statistical Analysis

Descriptive analyses were used to summarise the responses. The differences in quiz scores between subgroups were assessed using one-way ANOVA with Bonferroni post hoc tests, χ^2^ tests or Fisher’s exact as appropriate. Spearman’s correlation coefficients were used to assess the correlations between the respondents’ experience, self-reported confidence in diagnosis and quiz scores. The data analyses were performed with IBM SPSS 26.0 for Windows (IBM Corp. Armonk, NY, USA). A two-sided *p*-value of <0.05 was considered statistically significant in all analyses.

## 3. Results

### 3.1. Quiz Participants

A total number of 600 HCPs registered on the website; of those, 432 (72%) participated in the quiz. The participants’ demographics, profession, specialty and experience are summarised in Table 1.

Fifty-eight percent of the participants (n = 252) were from Europe, followed by Oceania (23%; n = 98) and America (8%; n = 98). The majority of participants were Caucasian/White (70%; n = 303) and 74% (n = 319) were medical clinicians. The participants represented healthcare professionals across specialties, including paediatrics (41%; n = 178), emergency medicine (21%; n = 90), primary care (14%; n = 60), emergency paediatrics (11%; n = 46) and dermatology (3%; n = 13). Approximately 60% (59%; n = 256) of participants had at least six years of experience since they qualified.

When asked about skin tone representation in participant’s educational resources during training, white (74%; n = 319) outnumbered mixed (25%; n = 106) and darker skin tones (2%; n = 7). The self-reported confidence in diagnosis was moderate. Only 5% (n = 21) of the participants were confident across a range of skin tones, 53% (n = 230) reported being “sometimes uncertain but clinically safe”, and 42% (n = 181) were generally uncertain.

### 3.2. Quiz Results

The distribution of the quiz scores is shown in Figure 2. The mean score was 5.37 with a standard deviation of 1.75, and the scores were normally distributed. Approximately 11% (n = 47) answered eight or more questions correctly. Of the 11% who passed, 62% (n = 29) were from Europe, 79% (n = 37) were Caucasian/White, 74% (n = 35) were medical clinicians and 61% (n = 29) had at least 6 years of clinical experience (Appendix A). With regards to clinical specialty, 34% (n = 16) were paediatricians, followed by primary care (21%; n = 10), emergency medicine (15%; n = 7), emergency paediatrics and dermatology (11%; n = 5). Their demographic background, clinical experience, training resources and self-reported confidence were similar compared to those who did not pass the quiz, except for the proportion of dermatologists (11% vs. 2%; *p* = 0.001).

### 3.3. Quiz Scores by Demographic and Professional Background

The quiz scores by subgroups are summarised in Figure 3. There was no significant difference amongst participants from different continents (*p* = 0.270), ethnic background (*p* = 0.397) or professions (*p* = 0.599).

There was a statistically significant difference observed in the mean quiz scores by specialty (*p* = 0.01). The post hoc comparisons identified a pair difference between dermatology (mean = 6.77; SD = 1.48) and other specialties, including paramedics, pharmacists and physiotherapists (mean = 4.73; SD = 1.95; Bonferroni post hoc test, *p* = 0.003).

The mean scores for the qualified HCPs ranged from 5.16 to 5.63. The students earned slightly lower scores in the quiz (mean = 4.69; SD = 1.74), but the differences across groups did not reach a significant level (*p* = 0.055).

### 3.4. Quiz Scores and Training Resources

The quiz scores did not differ significantly by skin tone representation in the participant’s training resources (*p* = 0.198). The mean score of the lighter skin tone was 5.44 (SD = 1.73) compared to a mix of skin tones, which was 5.23 (SD = 1.77), and darker skin tones, which was 4.43 (SD = 2.15). Further regrouping the variable into white/light vs. other skin tones did not change the conclusion (*p* = 0.171).

### 3.5. Quiz Scores and Self-Reported Confidence

The 21 participants who reported confidence in diagnoses across different skin tones earned a mean score of 5.62 (SD = 1.50), followed by those who were sometimes uncertain (mean = 5.45; SD = 1.74) and generally uncertain (mean = 5.24; SD = 1.78). The difference across the three groups was not significant (*p* = 0.400). The conclusion remained the same after further combining the two former groups (*p* = 0.242).

The correlation between experience, self-reported confidence and quiz score was also explored (Table 2). There was a weak correlation between experience and self-reported confidence (Spearman’s correlation coefficient ρ = 0.286; *p* < 0.01). No correlation was observed between confidence or experience with quiz scores (ρ = 0.087 and 0.076, respectively).

### 3.6. Well and Poorly Recognised Diagnoses

Four hundred and thirty-two participants answered 4320 questions covering 56 diagnoses (Appendix A). Among those, the cafe-au-lait macule was the most recognised diagnosis (95%), followed by scrofuloderma (94%), systemic lupus erythematosus (93%), chickenpox scars (90%) and staphylococcal abscess (85%) (Appendix A).

Poorly recognised diagnoses with less than 20% were signs caused by chemotherapy (19%), dog bite (17%), Kawasaki disease (15%), miliaria crystallina (11%), folliculitis (7%) and syphilis (6%). Of the 56 diagnoses in total, the study team picked 11 top diagnoses that were deemed to be the most common paediatric presentations and/or important to recognise (Figure 4).

## 4. Discussion

This is the first international study assessing the diagnostic knowledge of paediatric dermatological conditions across different skin tones by HCPs. These data show that the overall knowledge amongst HCPs in diagnosing paediatric dermatological conditions in darker skin tones was low, with a mean score of 50% and only 11% of participants scoring over the pass mark of 80%. There was no difference in score based on the participants geographical location, ethnicity or professional group. However, there was a statistically significant difference in specialty, with dermatologists scoring higher than others with a mean of 67%. Whilst this is expected, in the UK, children and young people will see their general practitioner or their emergency department practitioner first with their dermatological complaint, and so it is imperative that they are trained equitably to identify common paediatric skin conditions. A recent study looking at skin tone images within the top emergency medicine journals confirmed that images were more likely to be of light skin tones [14]. As the diagnosis of dermatological conditions is largely clinical, it is crucial that further training occurs for these front-line staff in order to improve their confidence and knowledge.

Perhaps most interestingly, these data highlight that there was no correlation between the diversity of training resources used and knowledge or self-reported confidence and knowledge. This was somewhat surprising given that our previous findings showed that the use of more inclusive resources positively correlated with increased confidence. This is reflected in the literature: a study looking at confidence in medical students before and after completing a dermatology education module with diverse skin tones showed that increased exposure to darker skin tones increased confidence in diagnosing conditions such as eczema (*p* = 0.006) and psoriasis (*p* = 0.028) [15]. Knowledge, however, was not objectively tested in our study. However, recent studies on equity and inclusion in medical education, training, access and research may easily support our hypothesis that there is not enough equity on this issue. Participation in clinical trials is historically imbalanced according to ethnicity [16,17,18,19,20], access in medical education [19,21] and access to care and diagnostics [21,22,23,24,25,26]. More specifically regarding paediatric dermatological conditions, it is well discussed the issue of the late recognition of jaundice in Black newborns and the disproportionate impact of kernicterus on this population [27]. Altogether, the mentioned evidence and our study point toward the same direction of a lack of enough attention to the equitable inclusion of diversity on every side of the medical sciences.

When looking at the top ten conditions, there was a surprising low identification of conditions that were deemed important to recognise in paediatrics. Only 40% of participants correctly identified eczema, although a larger proportion (66%) correctly identified infected eczema, perhaps due to the fact of seeing this presentation more frequently. Eczema is one of, if not the most common, skin conditions affecting more than 20% of children and young people nationally in the UK [28]. It has a large impact on the quality of life for children and young people and is associated with higher incidence of depression and anxiety, and so correct identification is crucial [29].

Meningococcal sepsis was recognised correctly in 61%. This is somewhat unsurprising given that the latest edition of the UK “Meningococcal Meningitis and Sepsis Guidance Notes” for General Practice, contains only two images of meningococcal septicaemia rash in children and young people with darker skin tones. There needs to be a shift in representation, especially in conditions such as meningococcal sepsis where there is evidence of a higher mortality in groups from ethnic minorities [30]. Kawasaki disease was the least recognised, with only 15% of participants identifying it correctly. The explanation for this may be that this rash is not pathognomonic, and so it is primarily diagnosed based on clinical history and correlation of features together.

## 5. Strength and Limitations

There were some limitations with the study, as the quiz was randomly generated and not identical for every participant which may have affected individual scores. This was a purposeful decision to avoid answers being shared within the DFTB online community. We acknowledge the relatively small sample size and unequal representation from non-White ethnicities. However, these are the largest data available on this topic, providing important data on the knowledge of clinicians in recognising childhood skin conditions.

## 6. Conclusions

These data highlight poor knowledge in diagnosing common dermatological conditions in children across all skin tones. In order to close the gap and ensure equity in patient care, it is crucial to improve the representation of all skin tones and facilitate teaching and training of common dermatological conditions in darker skin conditions.

## Figures and Tables

**Figure 1 children-09-01749-f001:**
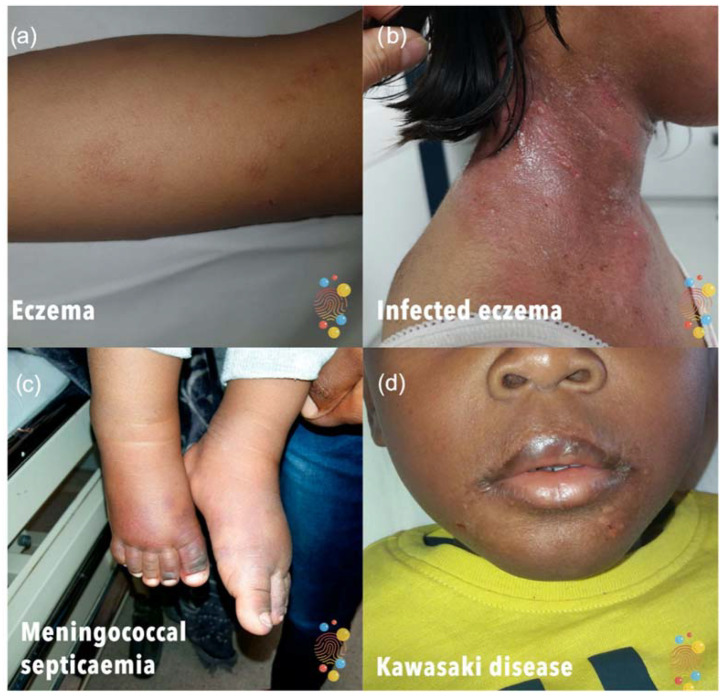
Example of images from the quiz. (**a**) Eczema; (**b**) Infected eczema; (**c**) Meningococcal septicalemia; (**d**) Kawasaki disease.

**Figure 2 children-09-01749-f002:**
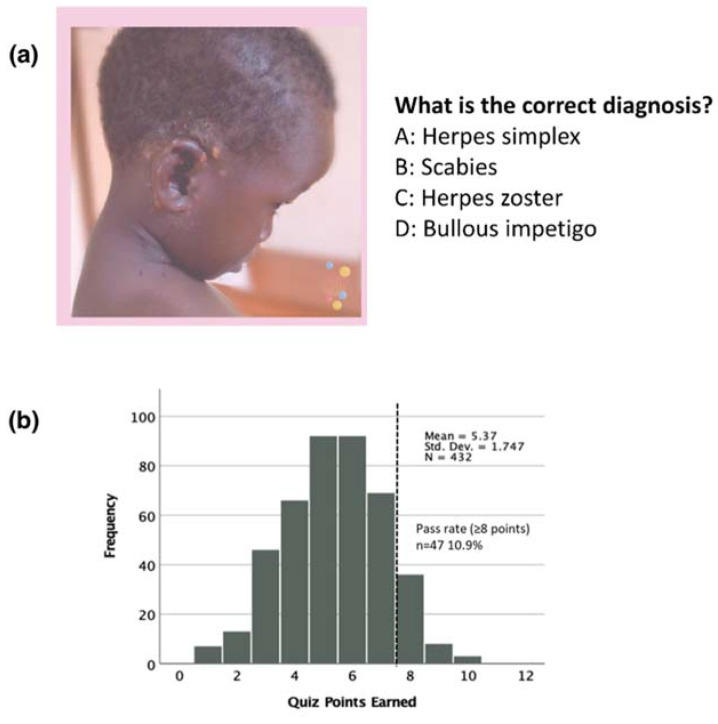
Example of a multiple choice question (**a**) and the distribution of the quiz scores (**b**).

**Figure 3 children-09-01749-f003:**
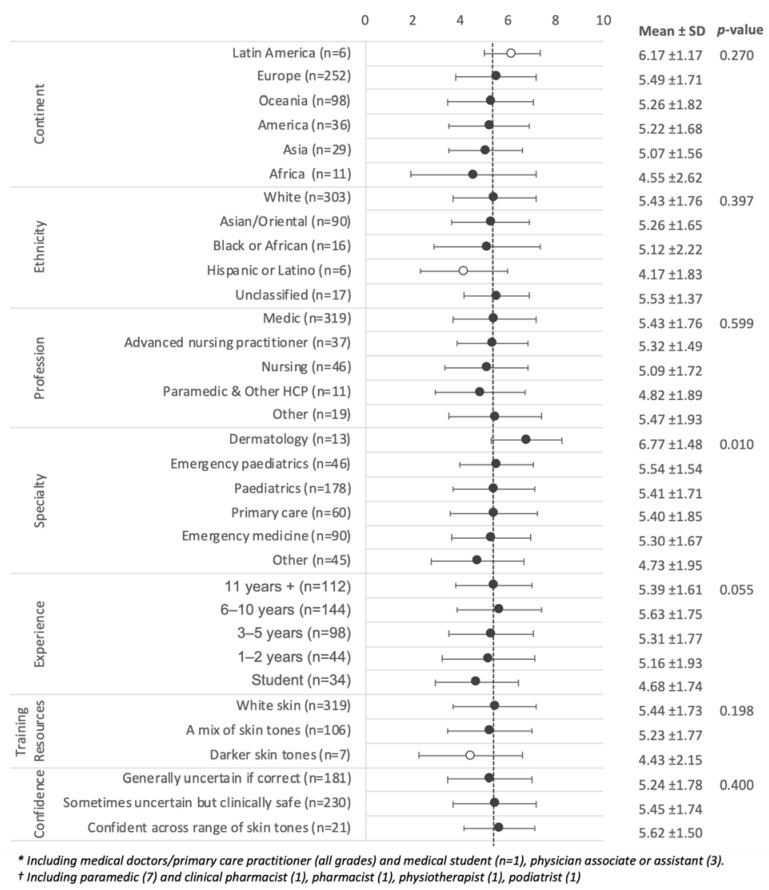
Quiz scores by demographics and professional background, experience, skin tone representation in training resources and self-reported confidence in diagnoses across a range of skin tones. The dotted line represents the group mean of 5.37, and open dots indicate subgroups with n < 10.

**Figure 4 children-09-01749-f004:**
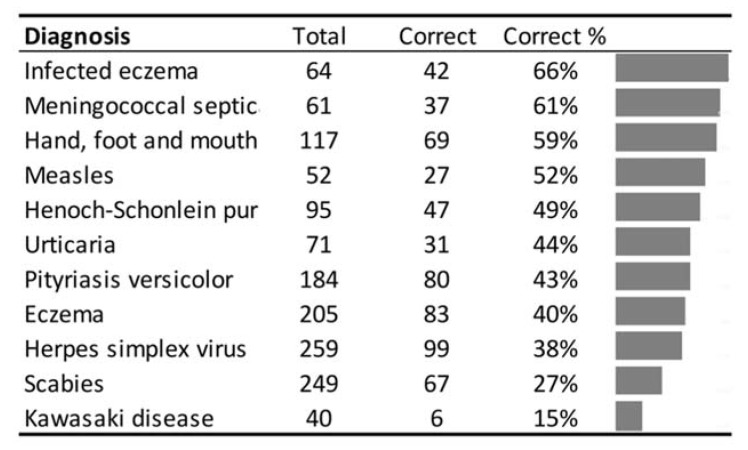
Percentages of correctness for common diagnoses.

**Table 1 children-09-01749-t001:** Demographics and characteristics of the study population (n = 432).

	n	Col%
**Continent**		
Europe	252	58%
Oceania	98	23%
America	36	8%
Asia	29	7%
Africa	11	3%
Latin America	6	1%
**Ethnicity**		
White	303	70%
Asian/Oriental	90	21%
Black or African	16	4%
Hispanic or Latino	6	1%
Unclassified	17	4%
**Profession**		
Medic *	319	74%
Nursing	46	11%
Advanced nursing practitioner	37	9%
Paramedic and other HCP	11	3%
Other (not specified)	19	4%
**Specialty**		
Paediatrics	178	41%
Emergency medicine	90	21%
Primary care	60	14%
Emergency paediatrics	46	11%
Dermatology	13	3%
Other ^†^	45	10%
**Experience**		
Student	34	8%
1–2 years	44	10%
3–5 years	98	23%
6–10 years	144	33%
11 years +	112	26%
**Majority Training Resources**		
White skin	319	74%
A mix of skin tones	106	25%
Darker skin tones	7	2%
**Confidence in Diagnoses**		
Generally uncertain if correct	181	42%
Sometimes uncertain but clinically safe	230	53%
Confident across range of skin tones	21	5%

* Includes medical doctors/primary care practitioners (all grades) and medical students (n = 1), physician associates or assistants (3). ^†^ Includes paramedics (7), clinical pharmacists (1), pharmacists (1), physiotherapists (1) and podiatrists (1).

**Table 2 children-09-01749-t002:** Correlation matrix between clinical experience, self-reported confidence in diagnosis and quiz score.

	Experience	Confidence	Quiz Score
**Experience**	1.000		
**Confidence**	0.286 **	1.000	
**Quiz score**	0.087	0.076	1.000

** Correlation is significant at the 0.01 level (2-tailed).

## Data Availability

Data sharing is available upon reasonable request from the corresponding authors.

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
