# Peer review of "Assessing Healthcare Professionals’ Identification of Paediatric Dermatological Conditions in Darker Skin Tones"

_children, 2022, doi:10.3390/children9111749_

Round 1
Reviewer 1 Report
Dear authors,
It is a well-written article, but it needs some arrangements.
- It need editing of English language gramer.
- The discussion part is very shallow, I expect it to be expanded.
Best wishes...
Author Response
Dear Editor and reviewer, thank you very much for your support. Please find below a point-by-point response to your comments. Changes have been highlighted in the manuscript as well
It is a well-written article, but it needs some arrangements.
Thanks for your appreciation and support
- It need editing of English language gramer.
English has been revised; all authors except the corresponding are native enligh speaker
- The discussion part is very shallow, I expect it to be expanded.
Thanks we agree and we did it, adding also about 10 references now
Best wishes...
Reviewer 2 Report
Nice job.
Author Response
Thanks a lot for appreciating our paper
Reviewer 3 Report
Assessing healthcare professionals’ identification of paediatric dermatological conditions in darker skin tones
The main aim of the presented article is assessing paediatric dermatological conditions in darker skin tones. The article consists of 7 (8 with References part) parts, including an abstract, an introduction, methods, results, discussion, strength and limitations and conclusion. In my opinion, the article is thoughtful and well-organized.
Abstract – I suggest adding the maximum number of points that could be scored (f.e. 5,37/10 - line 24).
Introduction - The concise introduction clearly indicates the problem of the work. I have to admit that I did not realize how big the problem is, how difficult it is to assess the skin of different shades (which even medical doctors pay attention to).
Methods – All work is based on the use of an online quiz with pediatric images of skin conditions in darker skin tones. I suggest to add the subtitle 2.1 Material and Methods or Study group and only then 2.2 Statistical analysis.
Results, Discussion and Conclusion – In my opinion the division of the Results part is a good idea. I have a few questions.
(1) Could the authors characterize the group (11%) that achieved a score of> 80%? Were they Europeans or not? Doctors what specialization? With what experience? Or were they just random people? It may be very interesting.
(2) I understand that white Europeans were the largest group and their result had the greatest impact on the average. On the other hand, there were no differences between continents and ethnicity. What causes that non-white races have problems with assessing dermatological disease in darker skinned patients? Can migration and learning in European centers have an impact on it? I suggest to add some comments to the Discussion part.
In each of paragraph of the Results we can see a short summary that has been presented in the form of a table, which is a repetition of the text, but it also helps the reader to organize the information.
Importantly, the authors are aware of the limitations of the work and clearly indicate them.
In my opinion, the results obtained in this experiment are terrifying in the context of diagnosis and treatment of the patient. For example, looking at the result of the recognition of eczema, which is “one of the commonest skin conditions affecting more than 20% of children and young people nationally”. It is comforting that the best result was obtained by dermatologists, although it was not a satisfactory result. Therefore, I think that the article is a much needed (especially in our time with such a huge migration of people of different races between all continents), helpful and important to show how the learning system should be changed/enriched.
Author Response
Dear Editor and Reviewe, thanks so much for your support aimed at improving our paper. Please find below a point-by-point resopnse to your comments. Changes have been highlighted in the manuscript
The main aim of the presented article is assessing paediatric dermatological conditions in darker skin tones. The article consists of 7 (8 with References part) parts, including an abstract, an introduction, methods, results, discussion, strength and limitations and conclusion. In my opinion, the article is thoughtful and well-organized.
Thanks so much
Abstract – I suggest adding the maximum number of points that could be scored (f.e. 5,37/10 - line 24).
Thanks, added
Introduction - The concise introduction clearly indicates the problem of the work. I have to admit that I did not realize how big the problem is, how difficult it is to assess the skin of different shades (which even medical doctors pay attention to).
Thanks
Methods – All work is based on the use of an online quiz with pediatric images of skin conditions in darker skin tones. I suggest to add the subtitle 2.1 Material and Methods or Study group and only then 2.2 Statistical analysis.
Done, thanks
Results, Discussion and Conclusion – In my opinion the division of the Results part is a good idea. I have a few questions.
(1) Could the authors characterize the group (11%) that achieved a score of> 80%? Were they Europeans or not? Doctors what specialization? With what experience? Or were they just random people? It may be very interesting.
(2) I understand that white Europeans were the largest group and their result had the greatest impact on the average. On the other hand, there were no differences between continents and ethnicity. What causes that non-white races have problems with assessing dermatological disease in darker skinned patients? Can migration and learning in European centers have an impact on it? I suggest to add some comments to the Discussion part.
Thanks, we have revised the result section extensively based on your comments
In each of paragraph of the Results we can see a short summary that has been presented in the form of a table, which is a repetition of the text, but it also helps the reader to organize the information.
thanks
Importantly, the authors are aware of the limitations of the work and clearly indicate them.
thanks
In my opinion, the results obtained in this experiment are terrifying in the context of diagnosis and treatment of the patient. For example, looking at the result of the recognition of eczema, which is “one of the commonest skin conditions affecting more than 20% of children and young people nationally”. It is comforting that the best result was obtained by dermatologists, although it was not a satisfactory result. Therefore, I think that the article is a much needed (especially in our time with such a huge migration of people of different races between all continents), helpful and important to show how the learning system should be changed/enriched.
thanks a lot for your support